# The DeepWater Horizon Oil Slick: Simulations of River Front Effects and Oil Droplet Size Distribution

**Lars Robert Hole** [1,*], **Knut-Frode Dagestad** [1,†], **Johannes Röhrs** [1,†], **Cecilie Wettre** [1,†], **Vassiliki H. Kourafalou** [2,†], **Yannis Androulidakis** [2,†], **Heesook Kang** [2,†], **Matthieu Le Hénaff** [3,4,†] and **Oscar Garcia-Pineda** [5,†]

[1] Norwegian Meteorological Institute, Allegt. 70, 5007 Bergen, Norway; knutfd@met.no (K.-F.D.); johannesro@met.no (J.R.); ceciliew@met.no (C.W.)
[2] Rosenstiel School of Marine and Atmospheric Science, University of Miami, Miami, FL 33149, USA; vkourafalou@miami.edu (V.H.K.); iandroulidakis@rsmas.miami.edu (Y.A.); hkang@rsmas.miami.edu (H.K.)
[3] Cooperative Institute for Marine and Atmospheric Studies, University of Miami, Miami, FL 33149, USA; mlehenaff@rsmas.miami.edu
[4] NOAA Atlantic Oceanographic and Meteorological Laboratory, Miami, FL 33149, USA
[5] WaterMapping, Gulf Breeze, FL 32563, USA; oscar.garcia@watermapping.com
[*] Correspondence: lrh@met.no; Tel.: +47-55236600
[†] These authors contributed equally to this work.

**Abstract:** The effect of river fronts on oil slick transport has been shown using high resolution forcing models and a fully fledged oil drift model, OpenOil. The model was used to simulate two periods of the 2010 DeepWater Horizon oil spill. Metocean forcing data were taken from the data-assimilative GoM-HYCOM 1/50° ocean model with realistic daily river input and global forecast products of wind and wave parameters from ECMWF. The simulations were initialized from satellite observations of the surface oil patch. The effect of using a newly developed parameterization for oil droplet size distribution was studied and compared to a traditional algorithm. Although the algorithms provide different distributions for a single wave breaking event, it was found that the net difference after long simulations is negligible, indicating that the outcome is robust regarding the choice of parameterization. The effect of removing the river outflow was investigated to showcase effects of river induced fronts on oil spreading. A consistent effect on the amount and location of stranded oil and a considerable impact on the location of the surface oil patch were found. During a period with large river outflow (20–27 May 2010), the total amount of stranded oil is reduced by about 50% in the simulation with no river input. The results compare well with satellite observations of the surface oil patch after simulating the surface oil patch drift for 7–8 days.

**Keywords:** HYCOM; OpenDrift; OpenOil; oil spill; modeling; simulations; satellite; observations; river fronts; DeepWater Horizon

## 1. Introduction

The synergy between shelf and open sea dynamics makes the Northern Gulf of Mexico (NGoM) a topographically and dynamically complex and interesting study area, in the presence of intense oil exploration [1,2]. Interactions of the Mississippi River (MR) plume and the Loop Current (LC) system were found to be important for the transport and fate of oil during the 2010 DeepWater Horizon (DWH) accident [3,4].

According to the U.S. Energy Information Administration, more than 45% of the total U.S. petroleum refining capacity and 51% of the total natural gas processing plant capacity are located

along the northern Gulf coast [5]. Oil leaks and accidents, such as the explosion on the DWH platform in 2010 (at 28.737° N, 88.366° W), have released significant quantities of hydrocarbons [6,7] in the sensitive marine environment around the MR Delta, and over the LouisianA TEXas (LATEX) and Mississippi Alabama FLoridA (MAFLA) shelves [3] (Figure 1).

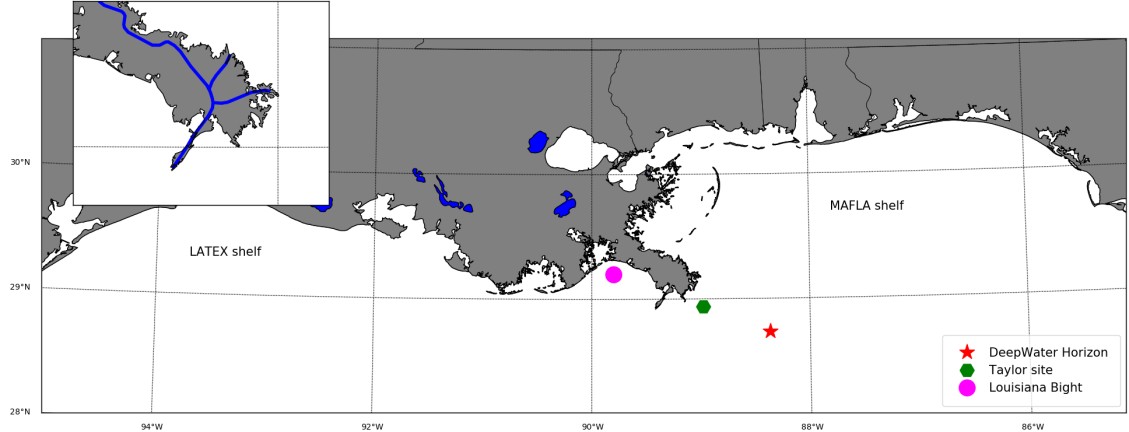

**Figure 1.** Map of the Northern Gulf of Mexico showing the geographical locations mentioned in the text. The map insert shows the Mississippi River (MR) delta in blue with the three major river passes that release MR water into the Gulf.

The present study was motivated by recent advances in the understanding of the role that salinity induced fronts and circulation areas dominated by river plume dynamics may play in the transport of hydrocarbons [3,8]. We used a comprehensive high resolution modeling system to demonstrate reliable estimates of hydrocarbon transport under the influence of all circulation driving mechanisms, with an emphasis on correctly accounting for fronts and circulation patterns due to river plume influence. Such fronts are density-driven, but also related to distinct circulation regimes that characterize river plume dynamics and may even control near-surface transport, subject to strong variability under the influence of additional shelf and deep sea flows. The river induced, buoyancy-driven flows include a westward coastal current along the LATEX shelf ("downstream" plume regime, i.e. in the direction of Kelvin-wave propagation) and a northeastward flow toward the MAFLA shelf ("upstream"). The former is due to the geostrophic balance between Coriolis and cross-shelf pressure gradient (e.g., [9–11]) and is enhanced by downwelling-favorable winds; such conditions favor material entrainment and nearshore confinement due to mass balance considerations. The latter is due to the balance between along-shelf pressure gradient and along-shelf acceleration [9]. This coastal flow has a tendency to turn offshore, enhanced by westerly upwelling-favorable winds; such conditions promote material removal away from the coastal zone, while they restrict material from offshore sources to reach the coasts. The third important flow regime, the buoyancy-driven anticyclonic bulge, is often suppressed due to the Mississippi Delta proximity to a steep slope ([12,13]) and it is usually formed under high discharge conditions [3]. These flow regimes induce well defined fronts that exhibit strong variability, depending on river discharge, winds and interaction with offshore flows [13]. These are relevant to pathways of any "materials", such as river-borne (e.g., nutrients) or from offshore sources potentially driven within plume influence (e.g., hydrocarbons from neighboring exploration sites). River induced fronts and related circulation regimes are hard to monitor and are often not well represented in numerical models. In the Gulf of Mexico (GoM), this is a particularly difficult task, due to complexity in topographic controls and direct influence of large scale oceanic currents (e.g., the LC and associated eddy field) on plume induced transport ([3,13]). Related transport pathways may have great impact on coastal ecosystems in the vicinity of GoM areas with river influence (especially over NGoM), but also remote ecosystems that can be reached along the LC (such as the South Florida islands

and reefs, e.g., [14–16]). In the simulations presented herein, we employed a detailed representation of river plume dynamics, thus the related circulation features and fronts are well described.

Many studies have dealt with simulations of the DWH spill [3,4,17–23] with focus on both subsurface [21] and surface [4] transport. North et al. [17] used a plume model to predict a stratification-dominated near field, in which small oil droplets detrained from the central plume containing faster rising large oil droplets and gas bubbles and became trapped by density stratification. They showed that simulated droplets with diameters between 10 and 50 μm formed a distinct subsurface plume, which was transported horizontally and remained in the subsurface for >1 month. In contrast, droplets with diameters >90 μm rose rapidly to the surface. Le Hénaff et al. [4] focused on oil transport on the water surface and showed that the wind played a major role on oil spreading over the NGoM. Barker [20] conducted Monte Carlo simulations consisting of 500 individual oil trajectory scenarios using historical data of water currents and winds. The results by Barker [20] indicate that, in approximately 75% of the scenarios, oil would be transported out of the GOM by the LC. This means that the actual trajectory of oil from the DWH falls in the 25% of scenarios.

Androulidakis et al. [8] carried out a field experiment deploying surface drifters at different times near the Taylor Energy Site, which is located in the vicinity of the MR outflow region over the NGoM and near the DWH site (approximately at 28.938° N, 88.978° W, Figure 1). This multi-platform observational experiment was conducted in April 2017 to investigate the main transport pathways from the Taylor Site either toward the NGoM continental shelves or offshore, toward the Gulf interior. The results indicate that the surface transport was determined by the MR plume extension over the Taylor Energy Site and the river induced fronts in combination with local circulation, prevailing winds and broader regional dynamics (LC system). The drifters, deployed during the field experiment in tandem with satellite data, drone imagery, wind measurements, and marine radar observations (surface currents and oil), described three main transport pathways, in agreement with the three prevailing circulation patterns of the MR plume [10,13,15,24].

The drifters followed the two prevailing coastal currents associated with MR plume dynamics (downstream/upstream moving westward/eastward of the Mississippi Delta) and an offshore pathway under the influence of basin-wide circulation (LC system). Near the Taylor site, the existence of multiple river fronts influenced the fate of oiled waters, preventing the transport of hydrocarbon toward the delta similar to a natural boom barrier, trapping and directing the oil either westward or eastward in agreement with Kourafalou and Androulidakis [3], who showed a similar interaction during the DWH accident using hydrodynamic numerical simulations. In situ thermohaline measurements around the Taylor Energy Site and across the river front showed that the MR plume near the Taylor Site was 5–10 m deep, while the clearer ocean water column was characterized by a 40 m upper-ocean homogeneous layer, mainly controlled by temperature [8]. Herein, we extend these observational and ocean modeling studies by conducting numerical simulations with a state-of-the-art oil spill model to investigate the river front effects on the evolution of the DWH oil spill in the summer of 2010.

Much progress has been made after the DWH accident in understanding the GoM ecosystem, the physical oceanography and its economic significance [1,25]. In this study, an open source Lagrangian oil drift model, OpenOil, was used to simulate the DWH oil spill evolution. OpenOil takes into account major factors that influence the short term drift of surface oil slicks such as metocean forcing (including Stokes drift), emulsification, evaporation, vertical entrainment and mixing. Simulations were initiated from satellite observations and a point source at the sea floor. The initialization of simulations from satellite observations is a relatively new feature in operational marine oil spill modeling [26], although pioneering work was carried out by Liu et al. [22] and Liu et al. [23] shortly after the DWH spill in 2010. The effect of two different oil droplet size distribution on the horizontal drift and vertical mixing is discussed. This study showcases how NGoM oil pathways are influenced by river plume dynamics and river induced fronts. Whether the use of realistic daily river discharge has a significant effect on the simulated location of the Surface Oil Patch (SOP) and stranding of oil was also investigated.

## 2. Materials and Methods

### 2.1. Shapefiles of Surface Oil Patch

Shape files derived from satellite analysis of the DWH SOP can be accessed through the NOAA-ERMA website [27]. In the present study, oil elements were seeded uniformly within the region enveloping the thick and thin oil slicks with no distinction. The shapefiles were used here for both initialization of the oil drift simulations and for verification of results.

### 2.2. Metocean Forcing

In the cases presented here, the ocean circulation fields came from a data-assimilative, high-resolution (1/50°, 1.8 km) configuration of the Hybrid Coordinate Ocean Model [28] in the GoM, developed by the authors (GoM-HYCOM 1/50). GoM-HYCOM 1/50 uses daily river forcing and data assimilation. The HYCOM model has a flexible, hybrid vertical coordinate system, in which the distribution of vertical layers is optimized: they are isopycnal in stratified water columns, terrain-following sigma in regions with sharp topography, and isobaric in the mixed layer and very shallow areas [29]. More information about the HYCOM model is available in the model user's manual ([28] and references therein). The GoM-HYCOM 1/50° covers the entire GoM and uses 32 vertical layers. This model configuration is similar to the one used by Le Hénaff and Kourafalou [30], with the realistic river forcing parameterization developed by Schiller and Kourafalou [10]. The river discharge data were obtained from the Army Corps of Engineers and the U.S. Geological Survey. The model is initialized in October 2009 with fields from the operational GLoBal HYCOM simulation run at the Naval Research Laboratory at the Stennis Space Center (GLB-HYCOM expt_90.8, [28]), and it is nested at the open boundaries with model fields from the same global simulation. The atmospheric forcing is based on the three-hourly winds, thermal forcing and precipitation forecast fields from the European Centre for Medium-Range Weather Forecasts [31], with spatial resolution of 0.125° (see below). The ECMWF provides daily global forecasts at 0:00 and 12:00 UTC with 0.125° resolution. Recent model upgrades have improved the overall performance of the forecasting system throughout the medium range. Further details on model description and verification can be found e.g., in the works of Ehard et al. [32] and Haiden et al. [33], and at ECM [31]. Here, ECMWF daily forecast products were used as atmospheric upper boundary conditions for the GoM-HYCOM 1/50 as well as for providing air temperature and wind drag for the OpenOil simulations with a 3 hourly time step.

The model assimilates satellite observations of Sea Surface Temperature and Sea Surface Height, and in situ observations of temperature and salinity from buoys, cruises, surface drifters, Argo floats and XBT casts. More details about the model configuration can be found in [16,24,30]. We used a detailed treatment of plume dynamics in HYCOM, based on Schiller and Kourafalou [10] that builds on a widely used parameterization that includes both salinity and momentum fluxes, pioneered by Kourafalou et al. [9]. A recent publication by Androulidakis et al. [16] shows the realism of the simulation employed here, with excellent agreement between modeled plume spreading and in situ data.

Two periods were studied here: 20–27 May 2010 and 2–10 July 2010. For the present study, altogether four simulations were performed, two for each period studied: one with the attributes mentioned above, called *Reference* simulation, and one called *no river*, in which the salinity fronts have been removed by shutting off the river discharge, setting precipitation to zero, and turning off the assimilation of salinity profiles. This is a procedure called "twin experiments" and is also used by other authors studying the effects of river fronts near large river outflows [34]. All other forcing conditions (e.g., meteorological and boundary) remained the same between the two experiments in order to investigate the impact of an individual forcing mechanism (here, the Mississippi buoyant discharge and the related density fronts) on the circulation features and furthermore on the oil spill extensions during the DwH period.

Wave properties were downloaded from the ECMWF third generation spectral WAve Model (WAM) global operational runs [35]. WAM is well known (see, e.g., [33,36]). The WAM model computes two-dimensional wave distribution, with 25 frequencies and 24 directions. The operational daily WAM forecasts used here are forced by the ECMWF atmospheric forecasts. WAM model output with 0.125° horizontal resolution was downloaded with 12-hourly time step and used here for estimating horizontal Stokes drift and vertical mixing of the oil with a three-hourly time step using linear interpolation.

From the two-dimensional wave spectra, the surface Stokes drift, significant wave height, and mean wave period were computed and used in the oil drift model for the calculation of wave-induced transport and mixing. For the wave-induced transport, a vertical Stokes drift profile is computed based on surface Stokes drift, significant wave height and mean wave period according to Breivik et al. [37].

According to the simulated wind conditions derived by the ECMWF, wind speed varied between 0.1 and 7.2 ms$^{-1}$ and the significant wave height varied between 0.1 and 1.2 m during the first period. In the second period, the corresponding values were 5–12 ms$^{-1}$ and 0.1–3.2 m. The various forcing data are summerized in Table 1.

**Table 1.** Summary of Metocean forcing used in the simulation.

| Model | Parameters | Resolution: Horizontal | Vertical | Temporal |
|---|---|---|---|---|
| NOAA shape files | initial location | | | |
| GOM Hycom | horizontal current | 1/50° | 32 layers | 3 h |
| ECMWF atmospheric model | wind velocity, air temperature | 1/8° | surface | 3 h |
| ECMWF wave model | Stokes drift, wave height and period | 1/8° | surface | 12 h |

### 2.3. The Oil Drift Model OpenOil

OpenOil is part of the the OpenDrift trajectory modeling framework [26], developed at the Norwegian Meteorological Institute and available as open source software from [38]. OpenOil has been evaluated against drifter and oil slick observations in the North Sea [39,40]. A similar wind drift factor is used in other oil spill trajectory modeling in GoM [41]. Details of the element tracking model are given in Dagestad et al. [26], and model physics that are specific to oil transport and fate are documented in Röhrs et al. [40] and in the following.

OpenOil is an integrated oil drift model consisting of sub-models for specific physical processes such as wave entrainment of oil [42], vertical mixing due to oceanic turbulence and waves [40,43], resurfacing of oil due to buoyancy [44], and emulsification taking account for oil properties [45]. The resurfacing is a function of oil density and droplet size following Stokes Law, and thereby the model physics are very sensitive to the specification of the oil's droplet size. Figure 2 shows the sequence of operations involved in the OpenOil simulations.

#### 2.3.1. Oil Droplet Size Distribution

Several algorithms are implemented to describe the oil's droplet size distribution, based on previous published parameterizations. The first option is based on the work of Delvigne and Sweeney [46] (DS88), manifesting a power-law droplet size number distribution as a function of droplet size, with an exponent of −2.3, i.e., there are many more small droplets than large droplets. Transferring this to a volume size distribution, as needed for practical oil spill simulation that follows the mass of the oil spill, the exponent becomes 0.7, i.e., there is more volume in the few large droplets than in the many small droplets. The typical droplet sizes range from 1 μm to 1 mm.

A second option to describe the droplet size distribution is based on Li et al. [47] (Li17), which takes the oil viscosity and the oil–water interfacial tension into account. This parameterization describes a log-normal law for the number size distribution, and the resulting volume size distribution

exhibits a peak at an intermediate droplet size of about $100\,\mu m$, depending on oil type and environmental conditions. Similar types of droplet size distribution have been developed and observed, confirming that there is a maximum in oil volume at a particular droplet size [48,49].

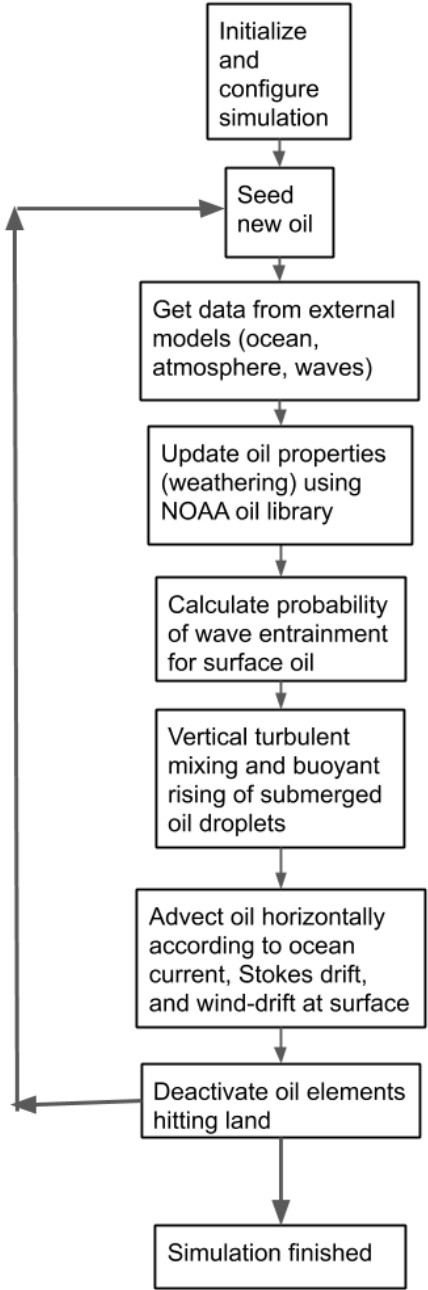

**Figure 2.** Flow chart showing the sequence of operations involved in the OpenOil simulations.

Following Li et al. [47], the volume (V) droplet size spectrum is described by the median droplet diameter, $D_{50}^{V}$, as

$$D_{50}^{V} = d_o r \left(1 + 10 \text{Oh}\right)^p \cdot \text{We}^q \qquad (1)$$

with the empirical coefficient $r = 1.791$ and the exponents $p = 0.460$ and $q = -0.518$. The PDF for the droplet size distribution follows a log-normal distribution around the medium diameter with a logarithmic base-10 standard deviation of $s = 0.38$ (Equation 16 in Röhrs et al. [40]).

The Weber number, We, is a dimensionless number describing the relative importance of inertial forces and oil–water interfacial tension. It is a function of the sea water density, $\rho_w$, the significant wave height, $H_s$, and the oil–water interfacial tension, $\sigma_{o-w}$, and is given by

$$\text{We} = \frac{\rho_w g H_s d_o}{\sigma_{o-w}},\tag{2}$$

where $g$ is the acceleration of gravity and $d_o = 4\sqrt{\frac{\sigma_{o-w}}{g(\rho_w-\rho_o)}}$ is the Rayleigh–Taylor instability maximum diameter.

The Ohnesorge number, Oh, is a dimensionless number describing the ratio of viscous forces to inertial and surface tension forces. It is a function of the dynamic oil viscosity, $\mu_o$, oil density, $\rho_o$, and oil–water interfacial tension:

$$\text{Oh} = \frac{\mu_o}{\sqrt{(\rho_o \sigma_{o-w} d_o)}}.\tag{3}$$

The volume size distribution, following Delvigne and Sweeney [46], is given by

$$V(d) = d^{-0.7}, d_{min} < d < d_{max}\tag{4}$$

where $d$ is the droplet diameter. Minimum and maximum droplet radii are set to $10^{-6}$ and $10^{-3}$ m, respectively. The exponent of $-0.7$ in the volume size distribution corresponds to an exponent in the number size distribution of 2.3 [44].

Droplet sizes are assigned to oil particles each time an element is submerged by breaking waves, following the wave entrainment algorithm of Li et al. [42]. The implementation of this algorithm in OpenOil is described with full detail in [40]. The droplet sizes for individual particles are drawn from a random distribution according to the chosen size distribution. The size distributions represent conditions for a stochastic wave entrainment event, representing equilibrium conditions during a model time step. It is noted that the overall size distribution of all submerged oil in the simulation is further subject to changes, as weather conditions, the oil's emulsification rate change and oil droplets of various sizes are subjected to various resurfacing time scales. Resurfaced elements are considered to be part of a surface slick, and are assigned a new droplet size distribution once they are re-entrained. Oil droplets at the sea surface (slick) are not considered to have a radius.

### 2.3.2. Droplet Size Distribution During Deep Blowouts

For oil elements released at the seafloor (wellhead), a simplistic and pragmatic approach of prescribing random radii in the range 0.5–5 mm is used, as suggested by Johansen [50].

### 2.3.3. Horizontal Transport

With regard to horizontal drift, three processes are considered: Any element, whether submerged or at the surface, drifts along with the ocean current. Elements are further subject to Stokes drift corresponding to their actual depth. Surface Stokes drift is normally obtained from a wave model, and its decline with depth is calculated as described in Breivik et al. [37]. Oil elements at the ocean surface are additionally moved with a factor of 2% of the wind. Together with the Stokes drift (typically 1.5% of the wind at the surface), which sums up to the commonly found empirical value of 3.5% of the wind speed [51]. The magnitude of the wind drift factor was discussed by Jones et al. [39] who stated that a 2% wind drift factor is required in OpenOil to reproduce their observations of a SOP in the North Sea. In essence, this is believed to be a compensation factor for the inability of any ocean model to represent the strong shear current in the upper few centimeters/decimeters of the ocean, and not surface oil actually moving relative to the water. It was found to be important in bringing the oil to the northern Gulf beaches [52,53].

The three horizontal drift components (currents, Stokes drift and wind drift) may lead to a very strong gradient of drift magnitude and direction in the upper few meters of the ocean. For this reason, it is also of critical importance to have a good description of the vertical oil transport processes.

### 2.3.4. Vertical Transport

Oil elements at the surface, regarded as being in the state of an oil slick, may be entrained into the ocean by breaking waves. The entrainment of oil droplets depends on both the wind and wave (breaking) conditions, but also on the oil properties, such as viscosity, density and oil–water interfacial tension. The buoyancy of droplets is calculated according to empirical relationships and the Stokes law following Tkalich and Chan [44], dependent on ocean stratification based on temperature and salinity from the ocean model, and the viscosities and densities of oil and water.

In addition to the wave induced entrainment, the oil elements are also subjected to vertical turbulence throughout the water column, described using a random-walk scheme based on the turbulent eddy diffusivity wind speed parameterization from Sundby [54].

### 2.3.5. Weathering

To calculate weathering of the oil, OpenOil interfaces with the open source ADIOS oil library [55], developed by NOAA and Lehr et al. [45]. In addition to state-of-the-art parametrization of weathering processes such as evaporation, emulsification and dispersion, this software contains a database of measured properties of almost 1000 oil types from around the world. As oils from different sources or wells have vastly different properties, such a database is of vital importance for accurate results. The ADIOS oil library is also used by the NOAA oil drift model [56].

The weathering algorithms describes evaporation and emulsification rate of oil, i.e., the water content. The emulsification and evaporation greatly affect oil density, viscosity and oil–water interfacial tension, and thereby the droplet size distribution through Equations (1)–(3). OpenOil takes into consideration weathering processes that are dominating in the initial oil spill period of 2–3 days. Long-term weathering processes such as sedimentation and microbial degradation are not considered in this study.

## 3. Results

A first set of simulations was carried out to investigate the effect of oil droplet size distribution. A second set focuses on the effect of river induced fronts. According to Crone and Tolstoy [6], the average flow rate from the oil well between 22 April and 3 June 2010 was estimated to 0.1 m$^3$ s$^{-1}$, assuming a liquid oil fraction of 0.4. Gas and highly volatile compounds are not considered here. After the riser was removed and until the leak was sealed on 15 July, the flow rate increased to 0.12 m$^3$ s$^{-1}$, corresponding to 10,368 m$^3$ day$^{-1}$. The amount of oil present at the sea surface at the initial time of the simulation was estimated based on the simulated removal rate, following the pioneering work of Liu et al. [22] and Liu et al. [23]. The residence time of oil at the sea surface depends heavily on oil properties as well as environmental conditions such as temperature of ocean and air, wind and waves, as described above. According to our mass balance calculations for the DWH spill, using the Light Louisiana Sweet oil type from the NOAA oil library and environmental conditions as described above, it seems reasonable to assume that 80% of the oil mass is removed from the surface after 10 days. This is within the range in our simulations that is typically 60–95% (see examples of mass balance plots further down). While Boufadel et al. [57] assumed a constant removal rate of surface oil 20% per day, the removal rate in the present simulation and in reality will vary with wind and wave conditions. The simulations for May 2010 were initialized by seeding 48,730 elements in a polygon obtained from NOAA shapefiles. Each element represents initially 1 m$^3$ oil. A continuous point source at the sea floor seeds an additional 8460 elements (8460 m$^{-3}$) per day during the simulation. After 3 June, these numbers were increased by 20% to 10,368 m$^3$ day$^{-1}$. The oil elements released at the surface were assigned droplet radii at each entrainment incident, according to the parameterization of

size distributions from respectively DS88 or Li17 (see [40] for details). Oil elements at the sea surface (slick) were not considered to have a radius.

Around 20–25 May 2010, there was a significant outflow of the Mississippi River (Figure 3) and part of the SOP was entrained along the LC resulting in a formation popularly referred to as the "tiger tail" (Figure 4) [53]. The OpenOil simulation shown in Figure 4a was carried out using the classical DS88 oil droplet size distribution [46]. Figure 4b shows the results from repeating this simulation using the new Li17 formulation. In Figure 5, the mass balance during seven days for the Li17 simulation is shown. There is virtually no difference between DS88 and Li17 and only Li17 is shown here. Both formulations result in about the same fraction of oil at the surface (about 50%) after seven days, with moderate wind speeds of up to 7.2 ms$^{-1}$. The light compounds evaporate fast after release, hence the more heavy compounds are tracked here ("dead oil"). Patches of thick oil (where the elements retain nearly 100% of their mass) are visible over a larger area (Figure 4). Larger oil droplets will rise more quickly to the surface [17,40], and DS88 provides a much higher fraction of large droplet after one hour compared to Li17 (Figure 6a,b). However, it turns out that the DS88 and Li17 provide similar volume distributions after a 24 h test simulation (Figure 6c,d). The peak in the distribution is still at around 100 μm for both formulations, and rapid rising to the surface can be expected according to North et al. [17]. Due to the small difference between DS88 and Li17, the DS88 simulation results in just marginally more oil stranded after seven days (13.2% vs. 12.8%), particularly west of the Mississippi Delta (Figure 4). A higher fraction of oil at the surface provides more efficient transport by wind and waves towards the shore and larger likelihood of stranding. For both simulations, the oil droplets quickly lose 40–50% of their mass, mostly through evaporation.

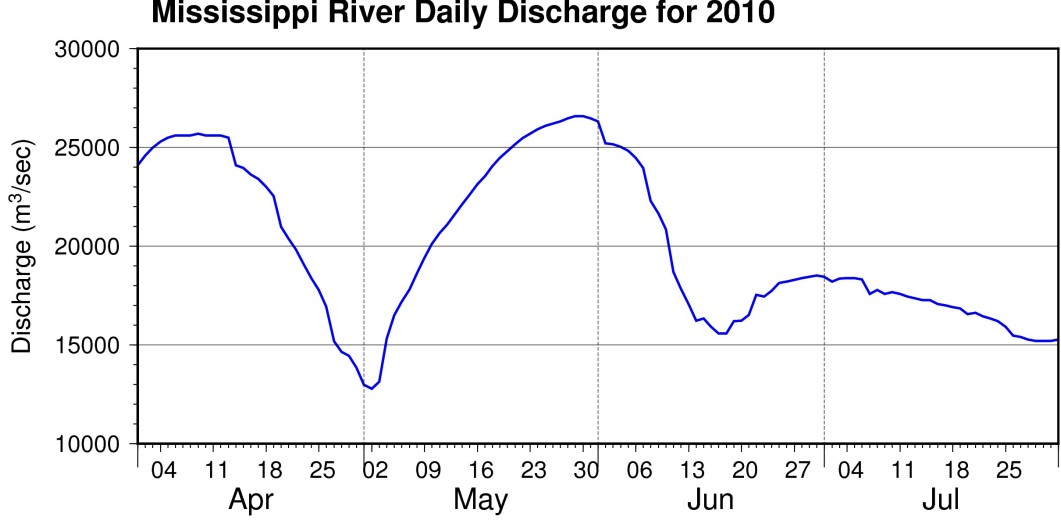

**Figure 3.** Discharge from Mississippi River in the Northern Gulf of Mexico during late spring and early summer 2010. Data kindly provided by U.S. Army Corps of Engineers.

Figure 7 shows the geographical distribution of droplet diameters at the surface. It is apparent that smaller droplets are present outside the MAFLA shelf, probably because the oil droplets have been subjected to more wind and hence wave action and natural dispersion in the last 24 h (Figure 8).

The Li17 formulation was applied for two types of simulations: *Reference* (all forcing data) and *no river* (all forcing data except for river runoff and precipitation). Two periods were chosen for these experiments: a high river discharge period with variable winds (20–27 May) and a relatively lower discharge period with persistent westward winds (2–10 July). The purpose was to investigate the effect of the salinity fronts by using the ocean circulation from the *no river* simulation, in which the precipitation and river discharge were turned off, while atmospheric and wave forcing were kept the same. Figures 9–12 show how the inclusion of river discharge in the ocean forcing can have opposite (and sometimes counter-intuitive) effects on oil transport. Table 2 summarizes the difference of the

*Reference* and *no river* simulations for 20–27 May, while Table 3 shows corresponding values for the 2–10 July simulations.

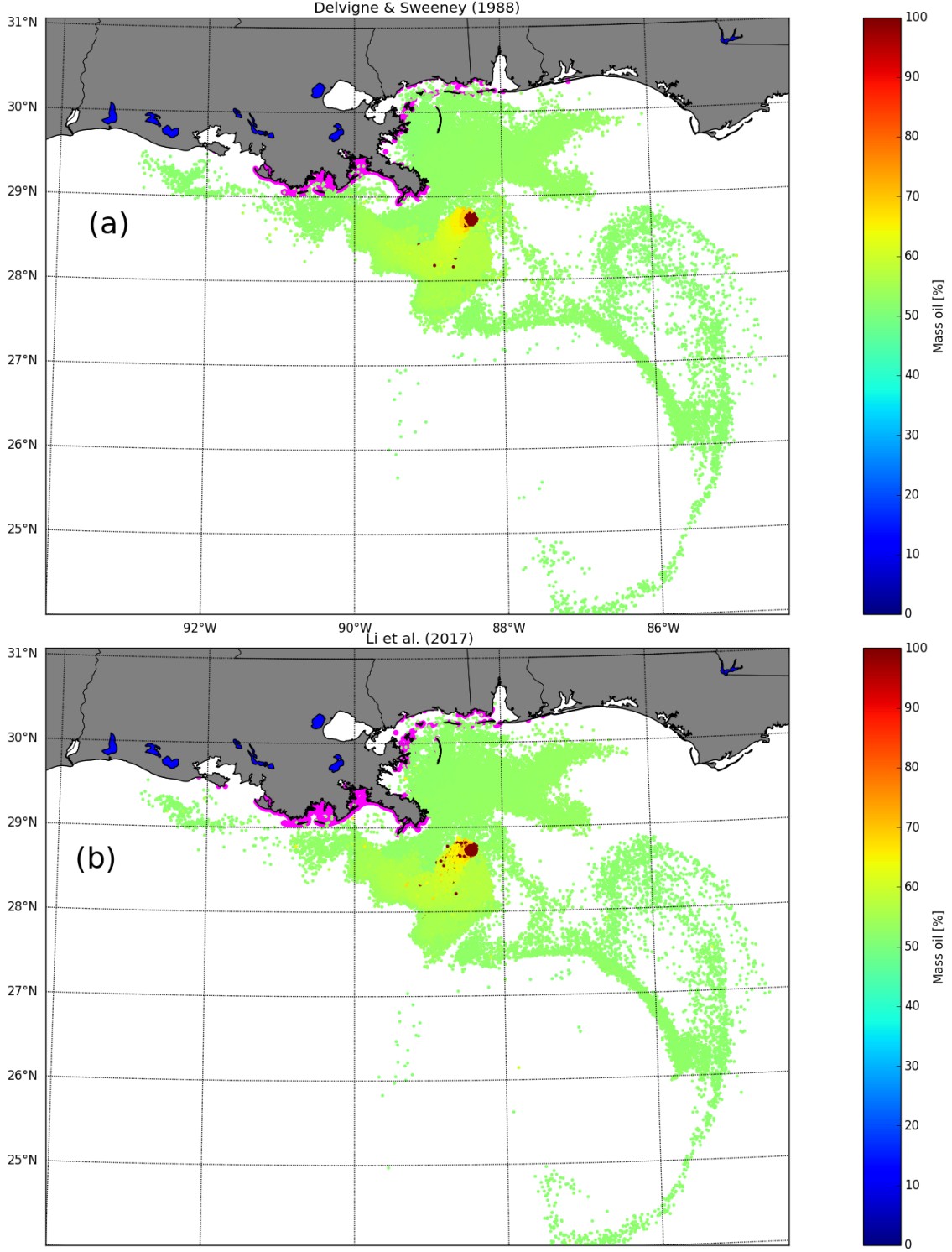

**Figure 4.** OpenOil simulation for 20–27 May 2010, using the Delvigne and Sweeney [46] oil droplet size distribution (**a**, 13.2% stranded oil), and the Li et al. [42] distribution (**b**, 12.8% stranded oil). Only surface oil elements are shown. Patch colors are the fraction of mass left in the elements. Magenta areas indicate stranded oil.

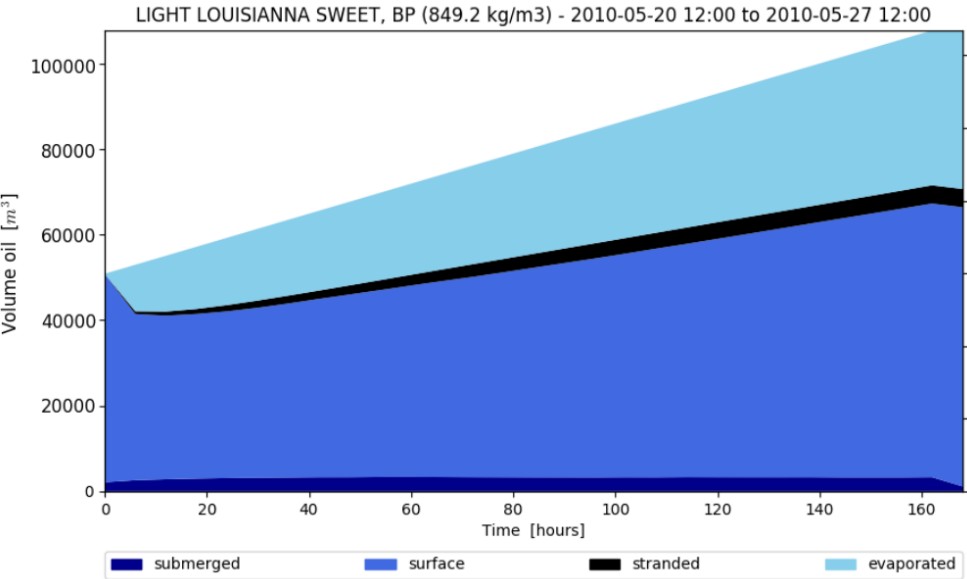

**Figure 5.** Mass balance of oil in the OpenOil simulation shown in Figure 2. The Li et al. [42] oil droplet size distribution is used.

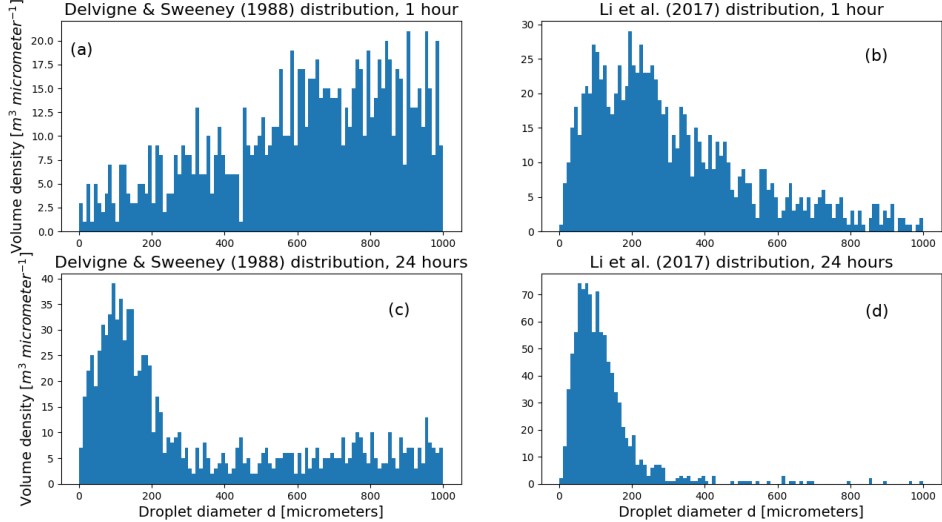

**Figure 6.** Oil droplet volume histogram for 1000 elements after 1 h using the *Light Louisiana Sweet, BP* oil type in OpenOil during 8 ms$^{-1}$ wind. The initial condition is a uniform distribution. Delvigne and Sweeney [46] (**a**) and Li et al. [42] (**b**). Bottom panels show corresponding distributions after 24 h.

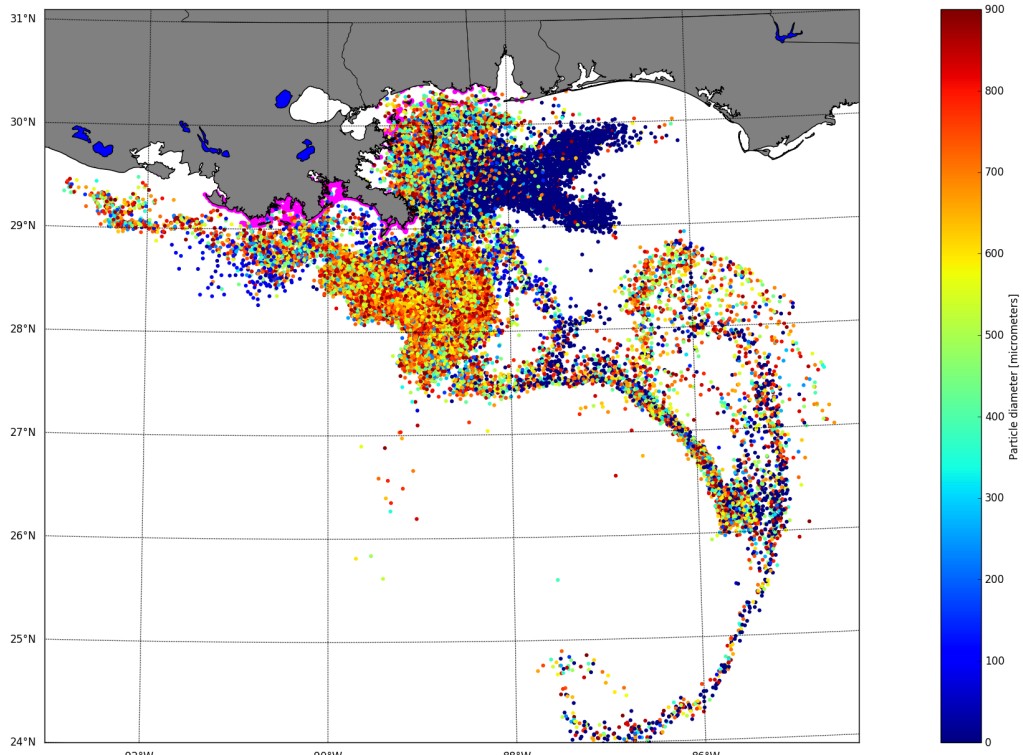

**Figure 7.** End condition of the *Reference* simulation 20–27 May 2010, showing all active elements (at surface and submerged). Same simulation as in Figure 4b. Stranded oil is shown in magenta. The color scale shows diameter of the oil droplets.

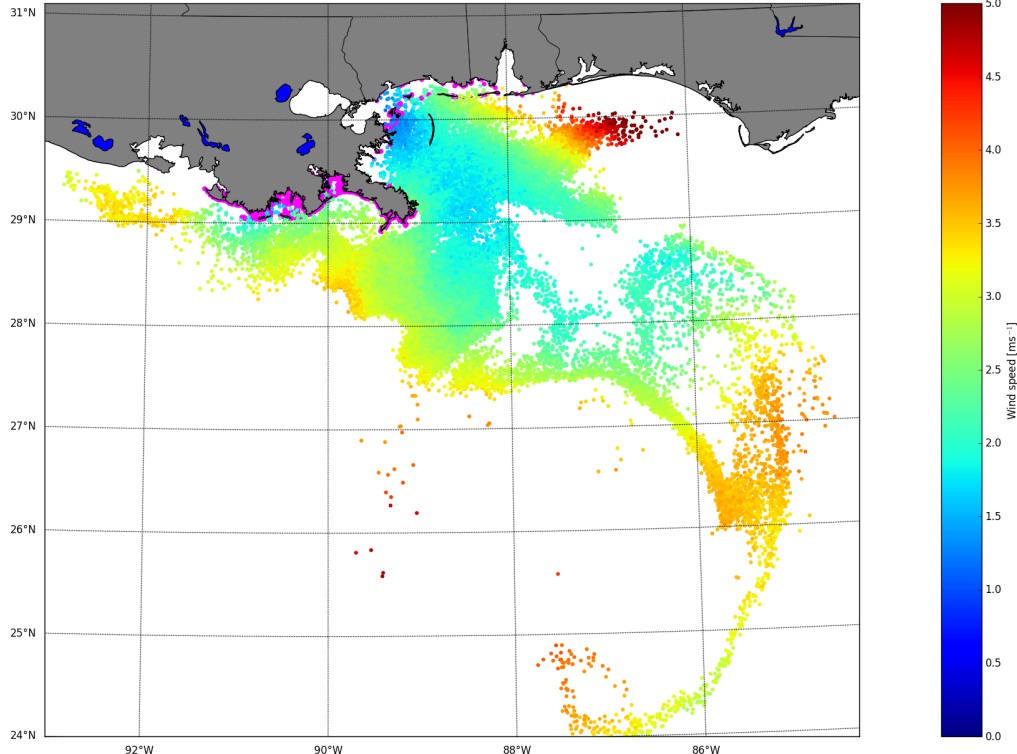

**Figure 8.** Same as Figure 7, but the color scale shows the average wind speed experienced by the element during the last 12 h.

**Table 2.** Percentages of stranded oil elements for the May 20–27 simulations.

|  | *Reference* Simulation | *no river* Simulation |
|---|---|---|
| West of MR Delta | 6.9 | 0.5 |
| MR Delta area | 4.0 | 3.4 |
| East of MR Delta | 1.2 | 2.1 |
| Total | 12.1 | 6.0 |

**Table 3.** Percentages of stranded oil elements for the July 2–10 simulations.

|  | *Reference* Simulation | *no river* Simulation |
|---|---|---|
| West of MR Delta | 10.6 | 7.8 |
| MR Delta area | 16.7 | 27.2 |
| East of MR Delta | 20.8 | 20.1 |
| Total | 48.1 | 55.1 |

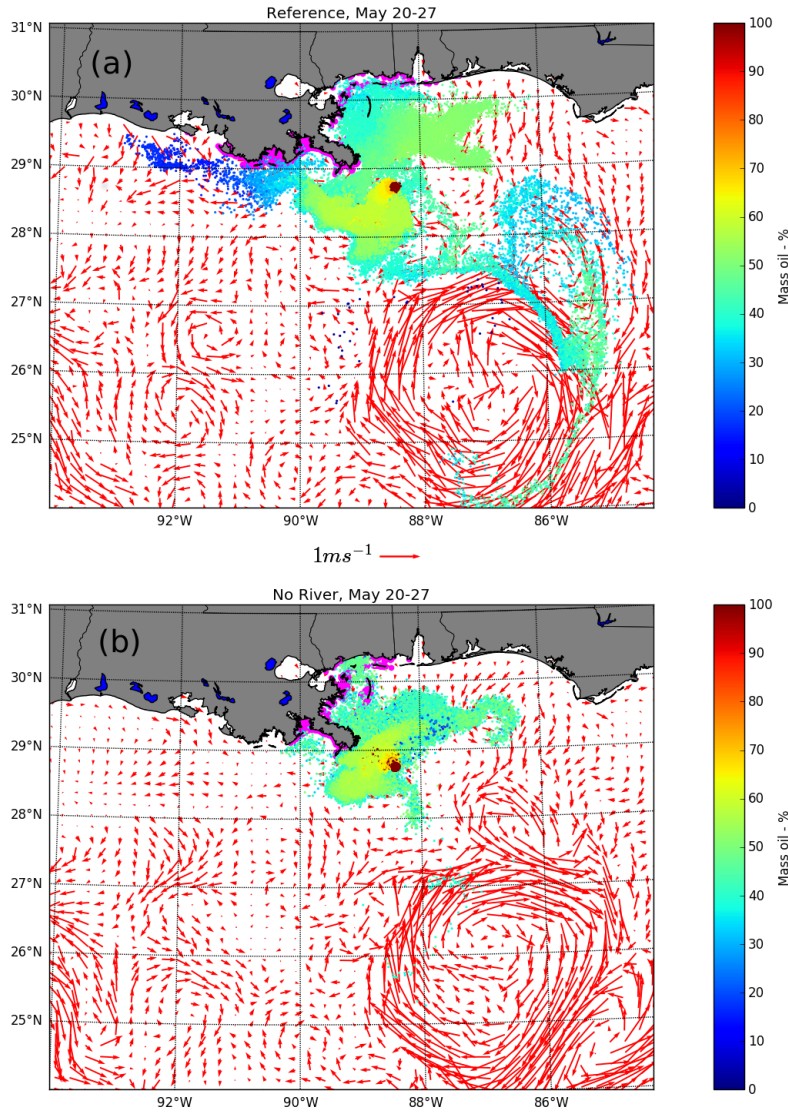

**Figure 9.** End condition of the simulation 20–27 May 2010, showing active (at surface) and stranded oil elements. Stranded oil is shown in magenta. The color scale indicate how much mass is left in each element. Red arrows are the GoM-HYCOM 1/50 forcing surface currents at the last time step of the simulation (every 5th data point shown). *Reference* simulation in (**a**) (12.1% stranded oil), and *no river* simulation shown in (**b**) (6% stranded oil) (see also Table 2).

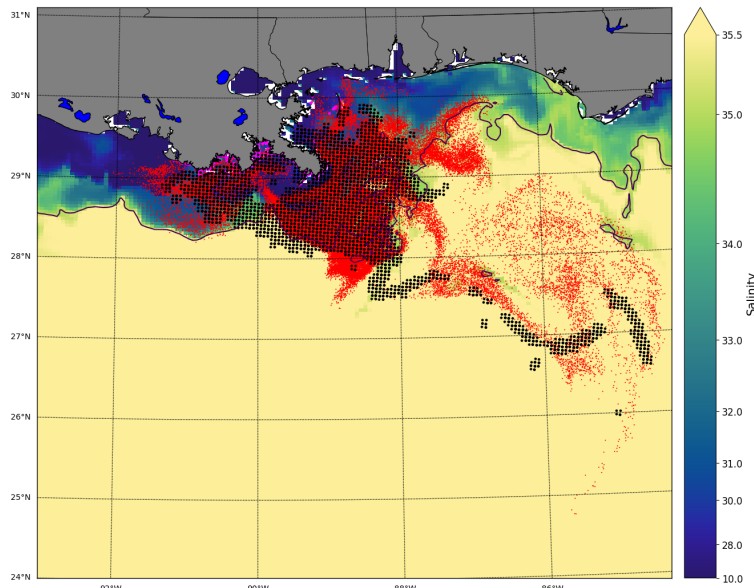

**Figure 10.** End condition of the *Reference* simulation 20–27 May 2010 same simulation as in Figure 9a, showing active elements at surface as red dots and the corresponding observed surface oil patch (NOAA shape file) as black dots. Modeled stranded oil is shown in magenta. The color scale shows sea surface salinity in the forcing data. The black contour is the 35 PSU isoline.

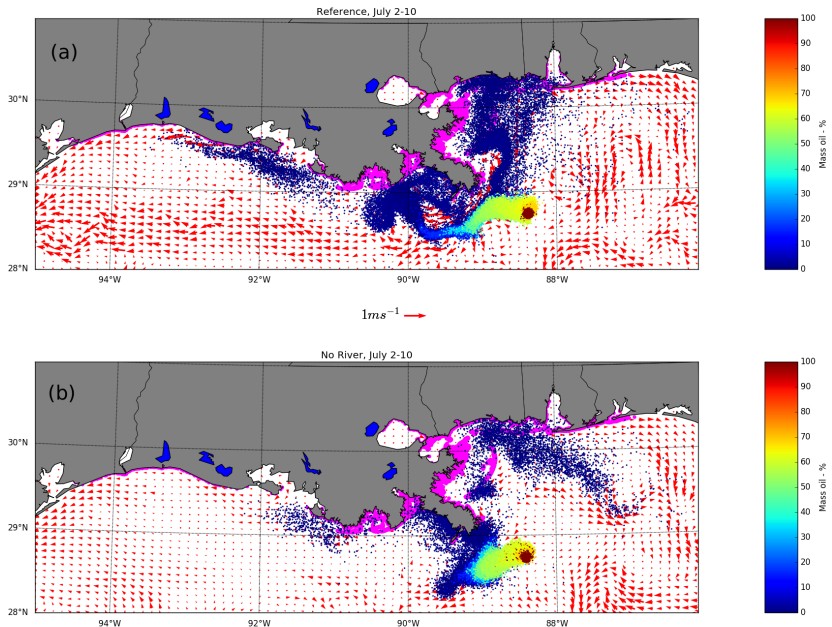

**Figure 11.** End condition of the simulation 2–10 July 2010, showing active (at surface) and stranded oil elements. Stranded oil is shown in magenta. The color scale indicate how much mass is left in each element. Red arrows are the GoM-HYCOM 1/50 surface currents at the last time step of the simulation (every 2nd data point shown). *Reference* simulation in (**a**) (48.1% stranded oil), and *no river* simulation in (**b**) (55.1% stranded oil) (see also Table 3).

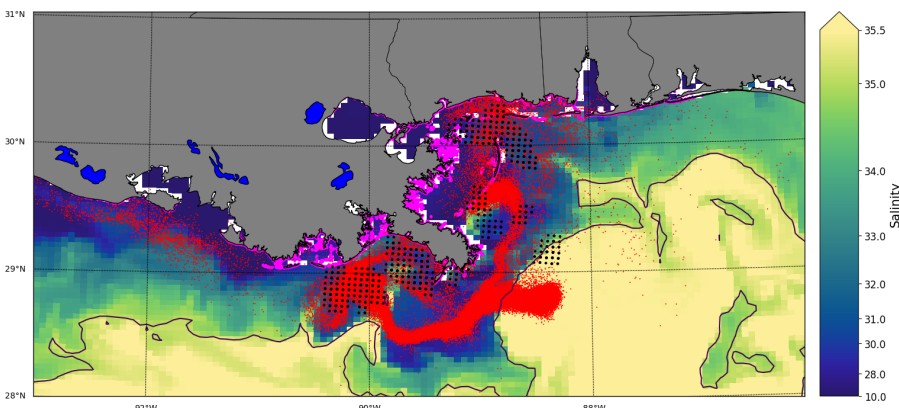

**Figure 12.** End condition of the *Reference* simulation 2–10 July 2010 (same simulation as in Figure 11b), showing active elements at surface as red dots and the corresponding observed surface oil patch (NOAA shape file) as black dots. Stranded oil is shown in magenta. The color scale shows sea surface salinity in the forcing data. The black contour is the 35 PSU isoline.

Kourafalou and Androulidakis [3], based on high-resolution ocean simulations over the NGoM, showed that the MR discharge peak around 20–30 May led to the formation of downstream (westward) and upstream (northeastward) plume areas that acted as a conduit for guiding oil toward the LATEX shelf and away from the MAFLA shelf, respectively (Figure 10). In the 20–27 May period, the river plume currents created a strong "bulge" that tended to turn waters clockwise around the Delta, with some waters moving westward. Figure 10 shows that the drift patterns of the SOP as observed in NOAA satellite images are generally reproduced by the model. Figure 10 also shows how the SOP is guided by the river fronts that stretches relatively far south in GoM. In addition, the offshore GoM circulation (LC and eddies) removed the riverine waters offshore, toward the GoM interior, forming the so-called "tiger tail" pathway. The removal of the MR input in the *no river* experiment during this period (Figure 9) allowed the spreading of oil toward the western MAFLA shelf due to the absence of the anticyclonic bulge; the stranded oil along the MAFLA coasts is more apparent in the *no river* case in comparison to the *Reference* experiment. In contrast, the amount of stranded oil is less along the western coasts (90 W–91 W) in the *no river* experiment due to the absence of the downstream MR plume pathway (Table 1). Moreover, the "tiger tail" signature is weaker in the *no river* simulation. Herein, we confirm with oil spill simulations that the strength of this MR offshore jet could have been an important factor in forming the "tiger tail" oil distribution pattern as also shown from observational data [58] and ocean simulations [3].

The second study period (2–10 July) was right after another high discharge period (although not as high as in May, Figure 3), promoting again a buoyancy-driven downstream current. This tendency was supported by downwelling-favorable winds [3], resulting in a clear westward transport of both low-salinity and oil containing waters, along a narrow band (of similar width) close to the LATEX coast and surrounding the MR Delta; extensive coastal areas of stranded oil are apparent along the western coasts in the *Reference* experiment (Figures 11 and 12). The removal of the river input (*no river* experiment) led to weaker downstream currents both close to the Delta (89.5° W) and along the western coasts (west of 90° W) and thus less stranded oil over the same region and more in the MR delta region (Table 3). The anticyclonic bulge, common in strong discharges and source of the downstream current [9], is completely absent in the *no river* experiment. As a result, more stranded oil was present closer to the Delta, inside the Louisiana Bight 3. It seems the absence of the anticyclonic bulge that was able to lead surface oiled waters directly west of the Louisiana Bight allowed the accumulation of oil very close to the Delta. In contrast, smaller differences between the two experiments were detected over the MAFLA region due to the weaker upstream currents during early July [3]. During this period of slightly higher wind speed, most elements appear to have lost 70–90% of their mass due to natural dispersion by waves and evaporation shortly after release (Figure 11).

## 4. Discussion

Several simulations of the DWH oil spill were carried out with high resolution forcing data and a Lagrangian oil spill model. Simulations were initialized from satellite observations of the SOP, and a continuous point source with a realistic spill rate at the sea floor.

Our results indicate that the two different formulations for oil droplet size distribution give similar results for both vertical and horizontal distribution of the oil, when wind speeds are typically 5–12 ms$^{-1}$ and breaking waves appear (Figure 4). Both formulations of the oil droplet size distribution result in the characteristic "tiger tail" shape of the SOP for the period 20–27 May 2010, and significant stranding in the delta west of the MR mouth in line with the observed SOP [3].

Both droplet size formulations that are used here (DS88 and Li17) result in similar size distributions after some time of simulation, as shown in Figure 6. Li17 prescribes a maximum droplet diameter in the volume distribution, as shown in Figure 6, which is due to two regimes in the size distribution where small droplets are limited by viscous effects, and larger droplets by oil–water interfacial tension [49]. This causes a peak in the volume distribution as seen in laboratory experiments with repeated mixing and wave breaking from the surface [49]. The DS88 distribution does not prescribe such a maximum, using a power-law that increases towards larger droplets in the volume distribution. However, the time-integrated simulations in OpenOil still produce a maximum in the droplet size distribution. The reason for this is the repeated wave breaking at the surface, which is more pertinent to large droplets that quickly rise to the surface. Hence, the description of buoyancy driven resurfacing and wave breaking in the oil spill model, together with the DS88 droplet size spectrum for individual wave breaking events, produces similar results to a more advanced droplet size distribution that explicitly prescribes a maximum in the volume size distribution. In conclusion, two very different droplet distributions give very similar distributions after some time. That indicates that the wave entrainment, vertical mixing and re-surfacing mechanisms that are part of OpenOil are more important for determining the final droplet size spectrum than the prescribed spectrum for individual wave breaking events.

A realistic description of droplet formation is required to describe the effects of an oil spill on the environment [17,40,57]. Figure 7 shows that the oil spill transport during the DWH spill favors a transport of small droplets towards the northeast, while larger oil droplets follow the paths towards southwest and southeast. As a result of their low buoyancy and turbulent mixing, smaller droplets are mixed into deeper parts of the ocean and subject to ocean currents at depth [40]. Larger droplets experience stronger buoyancy and are subject to surface currents while they may return to the surface slick. As wind and waves only affect the near-surface drift, the part of the oil slick that forms large droplets is quickly separated from the small droplets which retain at deep layers [17]. It was previously found that wave transport is important to bring oil the northern Gulf beaches [52]. The vertical mixing may also change the effect of the spill on the ecosystem: the parts of the oil spill at the surface is more hazardous to birds and the beach communities, while the small, submerged parts will have a substantially larger surface area to interact with water, fish and plankton [59,60].

The effect of realistic river discharge on the simulations was studied. One might expect that removing the river discharge would always bring the oil nearer to the shore, however interactions are complex. The *no river* simulation for 20–27 May showed more stranding oil, in particular close to the Louisiana Bight, but less stranding oil further downstream, along the LATEX shelf. The removal of the MR input reduced the downstream currents that were responsible for the westward transport of oiled waters along the LATEX shelf during the summer of 2010, as presented by Kourafalou and Androulidakis [3]. The MR plume and the accompanying river fronts were responsible to either entrap oil close to the coasts (e.g., LATEX shelf) or keep oiled waters offshore (e.g., MAFLA shelf) due to the formation of upstream currents (Figure 10). These results are in line with Kourafalou and Androulidakis [3] and the NOAA SOP observations used here and shown in Figure 9. It is also obvious that, in the *Reference* simulation, the oil elements are guided by the river fronts and they are carried further away from the coast, pushed into the LC south of 28° N and E of 88.5° W Figure 10.

The second simulation period (2–8 July) was right after a second high discharge period, promoting again a buoyancy-driven downstream current. This tendency is supported by downwelling-favorable winds, resulting in a clear westward transport of both low-salinity and oil-containing waters, along a narrow band close to the LATEX coast and surrounding the Mississippi Delta; extensive coastal areas of stranded oil are apparent along the western coasts in the *Reference* experiment. The removal of MR input (*No River* experiment) led to weaker downstream currents both close to the delta (89.5º W) and along the western coasts (west of 90º W) and thus less stranded oil over the same region.

The simulations presented here were initiated by seeding oil elements evenly in a polygon defined by NOAA satellite products [27], in addition to a continuous point source at the sea floor. The next possible step is to initiate the simulations from satellite products which contain information about oil film thickness in addition to area, and hence also quantify the amount of oil at the surface.

To the best of our knowledge, this is the first time that the importance of the effect of river fronts on oil slick transport in the Gulf of Mexico has been demonstrated using a fully fledged oil drift model nested on a high-resolution ocean model. Another new finding is that two very different oil droplet size distributions give similar net results compared to observed surface oil slicks. Future improvement of the OpenOil model could include implementation of horizontal dispersion by waves [61,62], but will be computationally demanding, and higher temporal resolution of the wave model output would be required. Bio-degradation is another factor important for longer simulations [21,63], which will be included in OpenOil in the near future.

Finally, as discussed by Liu et al. [23] and Liu et al. [53], human intervention activities to collect or remove the oil, for example, by use of dispersants, burning at sea, and skimming by boats, add uncertainties to the fate of the oil. This is not considered in the simulations presented here.

**Author Contributions:** Formal analysis, L.R.H., Y.A. and M.L.H.; Funding acquisition, V.H.K.; Project administration, V.H.K.; Software, K.-F.D., J.R. and H.K.; Writing—original draft, L.R.H.; and Writing—review and editing, J.R., C.W., V.H.K., Y.A., M.L.H. and O.G.-P.

**Funding:** This research was made possible by a grant from The Gulf of Mexico Research Initiative (award "Influence of river induced fronts on hydrocarbon transport", GOMA 23160700).

**Acknowledgments:** Atmospheric and wave data were kindly provided by the European Center for Medium-Range Weather Forecasts (ECMWF). M. Le Hénaff acknowledges partial support from the Physical Oceanography Division at NOAA's Atlantic Oceanographic and Meteorological Laboratory, AOML. This research was made possible by a grant from The Gulf of Mexico Research Initiative, award GOMA23160700. Data are publicly available through the Gulf of Mexico Research Initiative Information & Data Cooperative (GRIIDC) at https://data.gulfresearchinitiative.org, (GRIIDC - doi: 10.7266/N7NG4NPG, 10.7266/n7-gh86-8p66, 10.7266/n7-mw7c-bw15, and 10.7266/n7-11g0-cq20).

**Conflicts of Interest:** The authors declare no conflict of interest. The funders had no role in the design of the study; in the collection, analyses, or interpretation of data; in the writing of the manuscript, or in the decision to publish the results.

## Abbreviations

The following abbreviations are used in this manuscript:

| | |
|---|---|
| (N)GoM | (Northern) Gulf of Mexico |
| DWH | DeepWater Horizon |
| HYCOM | HYbrid Coordinate Ocean Model |
| ECMWF | European Center for Medium-Range Weather Forecasts |
| ADIOS | Automated Data Inquiry for Oil Spills |
| MR | Mississippi River |
| LC | Loop Current |
| LATEX | LouisianA TEXas shelf |
| MAFLA | Mississippi Alabama FLoridA shelf |
| NOAA | National Oceanic and Atmospheric Administration |
| USGS | US Geological Survey |

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
