# Peer review of "The DeepWater Horizon Oil Slick: Simulations of River Front Effects and Oil Droplet Size Distribution"

_jmse, doi:10.3390/jmse7100329_

Round 1

Reviewer 1 Report

Review of “The DeepWater Horizon oil slick: High resolution model simulations of river front effects, initialized and verified by satellite observations”

General comments:

River front effects on oil slick transport is demonstrated using an oil drift model. Also investigated is oil droplet sizes distribution. The work does provide some useful information to the oil spill community. However, the motivations of the work should be better discussed, because a relevant paper with similar topic (Influence of Mississippi River induced circulation on the Deepwater Horizon oil spill transport) has already been published by the co-authors. Some of the conclusions are not justified. Also, relevant papers are largely overlooked. I would suggest the manuscript be returned to the authors for some major revision before it can be considered for publication.

Major comments:

(1) The topic is on the effect of river fronts on oil slick transport. Actually, it is the ocean currents (and waves, etc) that transport oil slick. The difference between the experiments with and without Mississippi River (MR) is essentially the difference in simulating the relevant ocean currents in the region during that time. With the MR, the currents are more correctly simulated, while without the presence of the MR in the ocean circulation model, the currents would be quite different – may not be correctly simulated. This point should be properly clarified. Note that Kourafalou and Androulidakis (2013) have already published a nice paper on this topic. Don’t need to repeat the same topic (importance of the MR fronts on oil slick transport). What is new in this work?

(2) The title can be shorter. It should represent the new findings of this work. The effect of river fronts on oil slick transport is an old topic, which was published in Kourafalou and Androulidakis (2013). The initialization and verification by satellite observations are not the selling points of this paper (old topics, too).

(3) The abstract should be more concise.

(4) Where did you verify the model simulated oil locations with satellite observations? It would be good to show the verification maps side by side.

(5) It is concluded that the oil elements are guided by the river fronts and carried further away from the coast and pushed into the LC region (L390-391). But where are the river fronts? You only show the simulated oil end locations in Figure 10. Such trajectories could be formed without the presence of fronts. It’s necessary to add a figure to show the front locations to justify the statement.

(6) The authors state that “initialization of simulates from satellite observations is relatively new feature in marine oil spill modelling” and cite a new paper published in 2019 (L104-105). However, the technique was used for the first time in the rapid response to the DWH oil spill some 9 years ago. The relevant publications are largely overlooked.

2011, Tracking the Deepwater Horizon oil spill: A modeling perspective, Eos Trans., AGU, 92(6), 45-46, doi:10.1029/2011EO060001.

2011, Trajectory forecast as a rapid response to the Deepwater Horizon oil spill, in Monitoring and Modeling the Deepwater Horizon Oil Spill: A Record-Breaking Enterprise, Geophys. Monogr. Ser., 195, 153-165, doi:10.1029/2011GM001121.

Minor comments:

(1)  L66, the dedicated AGU book (Liu et al., 2011) should be cited here, as it includes several book chapters on the topic of SWH spill modeling.

2011, Monitoring and Modeling the Deepwater Horizon Oil Spill: A Record-Breaking Enterprise, Geophys. Monogr. Ser., 195, 271 PP., AGU/geopress, Washington, D.C., USA

(2)  L66, subsurface oil spill trajectory modeling, the early work in DWH (Weisberg et al. 2011) should also be cited.

2011, Tracking subsurface oil in the aftermath of the Deepwater Horizon well blowout, in Monitoring and Modeling the Deepwater Horizon Oil Spill: A Record-Breaking Enterprise, Geophys. Monogr. Ser, 195, 205-215, doi:10.1029/2011GM001131.

(3)  L104-105, the statements are wrong. The pioneering work of initializing oil spill trajectory model with satellite observations (Liu et al, 2011a, 2011b) should be properly credited here.

(4)  L115, here should add to the end of the sentence: “following the pioneering work of Liu et al. (2011b, 2011c)”.

(5) L158, “total sea” -- what do you mean here?

(6) L224, it would be good to add a sentence here: “It’s found to be important in bringing the oil to the northern Gulf beaches (Weisberg et al., 2017)”.

2017, On the movement of Deepwater Horizon Oil to northern Gulf beaches, Ocean Modell., 111, 81-97, http://dx.doi.org/10.1016/j.ocemod.2017.02.002.

(7) L228, it would be good to mention other group’s work here by adding a new sentence: “Similar wind drift factor is used in other oil spill trajectory modeling (Abascal et al., 2015).”

2015, Stochastic Lagrangian trajectory modeling of surface drifters deployed during the Deepwater Horizon oil spill, in Proc. 38th AMOP Tech. Seminar Env. Contamination & Response, Env. & Climate Change Canada, Ottawa, ON, PP. 71-99.

(8)  L272-273, how do you justify the removal rate’s variation with wind and wave condition? It would be good to clarify a fact by adding a new sentence: Human interventions, e.g., skimming and burning of surface oil and applications of dispersant, are not considered in the modeling.  These kind of processes post a challenge in accurate modeling of the DWH oil spill. This point has been discussed in Liu et al. (2011b, 2011c).

(9)  L319, the tiger tail pathway feature is the cover of the AGU book (Liu et al. 2011). It would be good to cite the book here.

(10)  L377, “at larger depths” should be changed to “at deep layers”.

(11)  L377, ”North et al. should be put in parentheses or completely removed from the sentence.

Author Response

General comments:

River front effects on oil slick transport is demonstrated using an oil drift model. Also investigated is oil droplet sizes distribution. The work does provide some useful information to the oil spill community. However, the motivations of the work should be better discussed, because a relevant paper with similar topic (Influence of Mississippi River induced circulation on the Deepwater Horizon oil spill transport) has already been published by the co-authors. Some of the conclusions are not justified. Also, relevant papers are largely overlooked. I would suggest the manuscript be returned to the authors for some major revision before it can be considered for publication.
We thank the reviewer for many constructive comments and suggestions for improvement of our manuscript.  Below are out point to point response to the reviewers remarks.

Major comments:

(1) The topic is on the effect of river fronts on oil slick transport. Actually, it is the ocean currents (and waves, etc) that transport oil slick. The difference between the experiments with and without Mississippi River (MR) is essentially the difference in simulating the relevant ocean currents in the region during that time. With the MR, the currents are more correctly simulated, while without the presence of the MR in the ocean circulation model, the currents would be quite different – may not be correctly simulated. This point should be properly clarified. Note that Kourafalou and Androulidakis(2013) have already published a nice paper on this topic. Don’t need to repeat the same topic (importance of the MR fronts on oil slick transport). What is new in this work?
We agree that ocean circulation is responsible for the oil transport. However, in both Kourafalou and Androulidakis (2013), based on hydrodynamic ocean simulations (not oil spill modeling), and Androulidakis et al. (2018), based on observations, we showed that river-induced fronts may alter the oil trajectories by trapping oil and direct it along the fronts under the effects of prevailing winds and surface currents. In the current study, we advance these findings by using oil simulations and develop twin experiments with and without river input. Moreover, we now investigate for the first time the impact of river-induced fronts on the amount and location of stranded oil. We believe that this study extends and clarifies the community’s knowledge about the overall role of the river plume during the most important phases of the DwH oil spill and may constitute one of the key reference works for future research in the field as Reviewer#3 mentioned. Moreover, the current study also focuses on other important topics such as the newly developed parameterization for oil droplet size distribution implemented in a state-of-the-art oil spill model (OpenOil). The new findings our now clarified in the revised manuscript.  Figures 10 and 12 are improved with contour lines to more clearly show the river fronts at the 35 PSU isolines (Androulidakis et al, 2019).

We also agree that the “no river” simulation does not provide the realistic circulation conditions, as it is missing an important circulation forcing mechanism for the study area (buoyancy due to river input). We now emphasize in the manuscript that this is an idealized simulation that is used to highlight the differences from the realistic simulation that are primarily due to the river input.

Androulidakis, Y., Kourafalou, V., Özgökmen, T., GarciaPineda, O., Lund, B., Le Hénaff, M., Hu, C., Haus, B.K., Novelli, G., Guigand, C. and Kang, H., 2018. Influence of RiverInduced Fronts on Hydrocarbon Transport: A Multiplatform Observational Study. Journal of Geophysical Research: Oceans, 123(5), pp.3259-3285.   

Kourafalou, V.H. and Androulidakis, Y.S., 2013. Influence of Mississippi River induced circulation on the Deepwater Horizon oil spill transport. Journal of Geophysical Research: Oceans, 118(8), pp.3823-3842.

(2) The title can be shorter. It should represent the new findings of this work. The effect of river fronts on oil slick transport is an old topic, which was published in Kourafalou and Androulidakis (2013). The initialization and verification by satellite observations are not the selling points of this paper (old topics, too).
The title has been shortened to “The DeepWater Horizon oil slick: simulations of river front effects and oil droplet size distribution”, which are the two topics we discuss in the manus. 

(3) The abstract should be more concise.
The abstract has been shortened considerably and is now hopefully more concise..

(4) Where did you verify the model simulated oil locations with satellite observations? It would be good to show the verification maps side by side.
The simulated locations are compared to NOAA satellite images (shapefiles) in Figs 10 and 12 together with salinity showing the fronts in the ocean. We have added extra sentences in the text to clarify this. 

(5) It is concluded that the oil elements are guided by the river fronts and carried further away from the coast and pushed into the LC region (L390-391). But where are the river fronts? You only show the simulated oil end locations in Figure 10. Such trajectories could be formed without the presence of fronts. It’s necessary to add a figure to show the front locations to justify the statement.
We agree with the reviewer that such trajectories such as the "tiger tail" can be formed without the presence of fronts. We do not imply that river fronts alone carry oil, but oil can be trapped together with the other material such as sargassum etc. along the density fronts, where surface currents inside the river plume may carry them along the shore or offshore. We clarify this now in the revised text. In a recent publication by Androulidakis et al. (2018), we demonstrated that oil can be trapped along the river fronts and move along these fronts either westward (downstream river currents) or northeastward (upstream river currents). Moreover, we also showed that the offshore spreading of the river plume currents and the respective density fronts may carry oil southward and away from the coast (observations related to the Taylor oil spill). In the current study, we demonstrate this type of effects in the DwH oil spill and advance the previous work by Kourafalou and Androulidakis (2013), based on oil spill simulations that were not used in our previous publications. We agree with the reviewer that front locations in Figure 10 are necessary and therefore we include them now in the revised manuscript.  

The river fronts can be seen as strong salinity gradients in Figs 10 and 12 and the 35 PSU isolines are now added. Also, please note that there was a 12 hour offset between the ocean model output and OpenOil output in the first version of this figure. We apologize for this error, which is now corrected.

(6) The authors state that “initialization of simulates from satellite observations is relatively newfeature in marine oil spill modelling” and cite a new paper published in 2019 (L104-105). However, the technique was used for the first time in the rapid response to the DWH oil spill some 9 years ago. The relevant publications are largely overlooked.

Liu, Y., R.H. Weisberg, C. Hu, and L. Zheng (2011), Tracking the Deepwater Horizon oil spill: A modeling perspective, Eos Trans., AGU, 92(6), 45-46, doi:10.1029/2011EO060001.

Liu, Y., R.H. Weisberg, C. Hu, and L. Zheng (2011), Trajectory forecast as a rapid response to the Deepwater Horizon oil spill, in Monitoring and Modeling the Deepwater Horizon Oil Spill: A Record- Breaking Enterprise, Geophys. Monogr. Ser., 195, 153-165, doi:10.1029/2011GM001121.
We agree that these publications should have been cited and they are now properly included. The first author must admit that he was not aware of this pioneering work. However, we now state the “The initialization of simulations from satellite observations is a relatively new feature in operational marine oil spill modelling”. 

Minor comments:

(1) L66, the dedicated AGU book (Liu et al., 2011) should be cited here, as it includes several book chapters on the topic of SWH spill modeling.

Liu, Y., A. MacFadyen, Z.-G. Ji, and R.H. Weisberg (Editors) (2011), Monitoring and Modeling theDeepwater Horizon Oil Spill: A Record-Breaking Enterprise, Geophys. Monogr. Ser., 195, 271 PP., AGU/geopress, Washington, D.C., USA(2) L66, subsurface oil spill trajectory modeling, the early work in DWH (Weisberg et al. 2011) should also be cited.

Weisberg, R.H., L. Zheng, and Y. Liu (2011), Tracking subsurface oil in the aftermath of the

Deepwater Horizon well blowout, in Monitoring and Modeling the Deepwater Horizon Oil Spill: A Record-Breaking Enterprise, Geophys. Monogr. Ser, 195, 205-215, doi:10.1029/2011GM001131.
These are now all properly cited.

(3) L104-105, the statements are wrong. The pioneering work of initializing oil spill trajectory model with satellite observations (Liu et al, 2011a, 2011b) should be properly credited here.
We agree that this is not correct. They are now properly cited.

(4) L115, here should add to the end of the sentence: “following the pioneering work of Liu et al. (2011b, 2011c)”.
This sentence is now included.

(5) L158, “total sea” -- what do you mean here?
This information is not relevant here and the sentence is shortened. We use the combined effect of wind sea and swell.

(6) L224, it would be good to add a sentence here: “It’s found to be important in bringing the oil to the northern Gulf beaches (Weisberg et al., 2017)”.

Weisberg, R.H., L. Zheng, and Y. Liu (2017), On the movement of Deepwater Horizon Oil to northern Gulf beaches, Ocean Modell., 111, 81-97, http://dx.doi.org/10.1016/j.ocemod.2017.02.002.
We agree, and this sentence and reference is now included.

(7) L228, it would be good to mention other group’s work here by adding a new sentence: “Similar wind drift factor is used in other oil spill trajectory modeling (Abascal et al., 2015).”

Abascal, A.J., S. Castanedo, R. Minguez, R. Medina, Y. Liu, and R.H. Weisberg (2015), Stochastic Lagrangian trajectory modeling of surface drifters deployed during the Deepwater Horizon oil spill, in Proc. 38th AMOP Tech. Seminar Env. Contamination & Response, Env. & Climate Change Canada, Ottawa, ON, PP. 71-99.
We agree. This sentence and reference is now included in the “Horizontal transport” section where the wind drift factor is discussed.

(8) L272-273, how do you justify the removal rate’s variation with wind and wave condition? It would be good to clarify a fact by adding a new sentence: Human interventions, e.g., skimming and burning of surface oil and applications of dispersant, are not considered in the modeling. These kind of processes post a challenge in accurate modeling of the DWH oil spill. This point has been discussed in Liu et al. (2011b, 2011c).
The removal rate used to calculate the amount of oil at the surface is our best estimate from the calculated mass balance (see e.g. Fig 5), but of course it will vary with time. We to have added some discussion on the uncertainty related to human intervention and refer to Liu et al for further detail. 

(9) L319, the tiger tail pathway feature is the cover of the AGU book (Liu et al. 2011). It would be good to cite the book here.
We agree, this is done now.

(10) L377, “at larger depths” should be changed to “at deep layers”.
We agree, this is done now.

(11) L377, ”North et al. should be put in parentheses or completely removed from the sentence.
We agree. It is put in parentheses now.

Reviewer 2 Report

The manuscript “The DeepWater Horizon oil slick: High resolution model simulations of river front effects, initialized and verified by satellite observations” analyzes the results of model simulations of river front effects on The DeepWater Horizon oil slick oil, for the period of catastrophic oil discharge in 2010.

Despite the fact that many scientific publications are devoted to this disaster, the authors presented new and interesting results.

The manuscript is clearly presented and describes the methodology with sufficient details.

The manuscript is acceptable for publication but, I would like to make a small remark:

Words “verified by satellite observations “ are present in the title of the manuscript . A lot of satellite data was obtained for the period considered in the article, in particular the twentieth of May. During this period, a tail or stream of oil pollution was observed. The authors also received a “tail” as a result of modeling, but it has a completely different direction than that observed in satellite images. I would like the authors to provide an explanation for this discrepancy.

Author Response

The manuscript “The DeepWater Horizon oil slick: High resolution model simulations of river front effects, initialized and verified by satellite observations” analyzes the results of model simulations of river front effects on The DeepWater Horizon oil slick oil, for the period of catastrophic oil discharge in 2010.

Despite the fact that many scientific publications are devoted to this disaster, the authors presented new and interesting results.

The manuscript is clearly presented and describes the methodology with sufficient details.

The manuscript is acceptable for publication but, I would like to make a small remark:

Words “verified by satellite observations “ are present in the title of the manuscript . A lot of satellite data was obtained for the period considered in the article, in particular the twentieth of May. During this period, a tail or stream of oil pollution was observed. The authors also received a “tail” as a result of modeling, but it has a completely different direction than that observed in satellite images. I would like the authors to provide an explanation for this discrepancy.

We agree that oil advected along the tiger tail obtained from the Reference model simulation (after a one week simulaiton) shows some differences with respect to observations. In particular, some oil particles are entrained anticlockwise in a cyclonic eddy around (27.5°N; 86°W), which is seen only partially in the observations with the branch going northeastward at (27°N; 86°W). Moreover, some oil particles are entrained clockwise around the Loop Current, which is not seen on the observations on May 27 (Figure 8) but is visible on observations on May 30 (see e.g. Liu et al., 2011b, their Figure 5). This pattern is thus realistic, although it was observed with a 2-3 day delay when compared to the simulation outputs. In addition, both patterns are composed of very few oil particles, which represent only small quantities of oil. These discrepancies are likely due to the sensitivity of Lagrangian particle tracks along fronts to initial conditions and to ocean dynamical features, as a small mismatch in the initial location of the oil or in the location and extend of ocean eddies and fronts can lead to the rapid advection of the simulated oil to areas outside the location where oil was observed. In spite of those small discrepancies, the main branch of the tiger tail, which is directed southeastward between 26°N and 28°N and presents a high density of oil particles, has a realistic extension in the oil simulation.

Reviewer 3 Report

The paper deals with the simulation of the DWH oil spill, and is  the product of intensive work carried out with the most up-to-date methodology, both numerical and experimental. The discussion of the physical problems is extensive and accurate.

The results are relevant not only to the specific episode, but also to the more general problem of oil slicks in the sea. It may constitute one of the key reference works for future research in the field.

The presentation is very clear, and in my opinion, the paper certainly deserves publication on J. Mar. Sci. Eng.

In the following, a few suggestions for the Authors which might improve the appeal of the work. Some of the remarks are also reported on the accompanying pdf file.

line 47 can you spend a word or two to clarify why downwelling should cause nearshore confinement? It is not intuitive, at least not to me.

line 132 I would be interesting to understand what are the effects of the river flow input on the  HYCOM results. Is  just the mass balance and salinity /temperature equations that are affected or also the momentum? In most situations I know of , the river flow momentum input is negligible.

line 147 and 154. Why didn’t you use analysis or re-analsysis data (rather than forecast)?

line 162 This part could be made clearer. I assume you take ECMWF wave data to compute drift ; is the computation carried out  by OpenOil? and what does OpenOil take as input? Not the full spectrum, I believe,, so probably just the total SWH as I gather from the reference Breivik et al that you quote in the following. So, what is the use of the separation between  Wind and Swell you mentioned earlier? CMWF WAM data do supply that information, but is it of any use to you?

line 221 to 231

 How is the computation of the wave drift actually carried out?  See also remark above, at line 162

You mention a 2% wind  speed velocity additional factor, beside the computed Stokes drift,  to reproduce the observations.. Is it your case too? Did you use this additional factor  to fit your data? This is an important aspect, because, as your reference (Jones et al) quite rightly states, the mechanism behind this additional direct   is not clear.  Any information to this regard would be very useful to the readers.

And

 have you considered also the possibility that dispersion (as opposed to net transport ) due to wave drift might play a role? Indeed the effects of wave drift not only determine  the position of a spill but also influence its spreading due to the non to the angular spreading of thewave spectrum

 The following two papers might be useful:

Carratelli, E.P., Dentale, F., Reale, F. On the effects of wave-induced drift and dispersion in the deepwater horizon oil spill(2011) Geophysical Monograph Series, 195, pp. 197-204.

Giarrusso, C.C., Pugliese Carratelli, E., Spulsi, G. On the effects of wave drift on the dispersion of floating pollutants (2001) Ocean Engineering, 28 (10), pp. 1339-1348.

line 232 what do you mean by “three components of the horizontal drift “? Can you explain?

Author Response

The paper deals with the simulation of the DWH oil spill, and is the product of intensive work carried out with the most up-to-date methodology, both numerical and experimental. The discussion of the physical problems is extensive and accurate.

The results are relevant not only to the specific episode, but also to the more general problem of oil slicks in the sea. It may constitute one of the key reference works for future research in the field. The presentation is very clear, and in my opinion, the paper certainly deserves publication on J. Mar. Sci. Eng. In the following, a few suggestions for the Authors which might improve the appeal of the work. Some of the remarks are also reported on the accompanying pdf file.

line 47 can you spend a word or two to clarify why downwelling should cause nearshore confinement? It is not intuitive, at least not to me.

We thank the reviewer for very positive remarks. We believe that downwelling will cause onshore transport due to mass balance consideraitons and hence nearshore confinement. We added a sentence describing this. 

line 132 I would be interesting to understand what are the effects of the river flow input on the HYCOM results. Is just the mass balance and salinity /temperature equations that are affected or also the momentum? In most situations I know of , the river flow momentum input is negligible. 

We use a detailed treatment of plume dynamics in HYCOM, based on Schiller and Kourafalou (2010) that builds on a widely used parameterization that includes both salinity and momentum fluxes, pioneered in Kourafalou et al. (1996). Our approach is to have a robust underlying plume dynamics that are in close agreement with a theoretical approach (rather than a simple salt flux that many modelers employ for river representation). The impact of momentum fluxes can be best quantified in idealized experiments, with simplified external forcing. In studies with full forcing, as the present, this impact can vary during different forcing conditions. However, the parameterization that we have adopted from Schiller and Kourafalou (2010) is undoubtedly superior to the initial one in HYCOM (as a virtual salt flux, added in the precipitation field, see Huang 1993) and allows a much more realistic plume development and spreading of the riverine waters. This parameterization has been included in subsequent releases of the "official" HYCOM code, as a user choice. A recent publication by Androulidakis et al. (2019) shows the realism of the simulation employed here, with excellent agreement between modeled plume spreading and in situ data. 

Huang, R.X., 1993. Real freshwater flux as a natural boundary condition for the salinity balance and thermohaline circulation forced by evaporation and precipitation. Journal of Physical Oceanography, 23(11), pp.2428-2446.

Schiller, R.V. and Kourafalou, V.H., 2010. Modeling river plume dynamics with the HYbrid Coordinate Ocean Model. Ocean Modelling, 33(1-2), pp.101-117.

Androulidakis, Y.S., V.H. Kourafalou, M. Le Hénaff, H. Kang, T. Sutton, S. Chen, C. Hu, N. Ntaganou, 2019. Offshore spreading of Mississippi waters: pathways and vertical structure under eddy influence. J. Geophys. Res., 124, doi: 10.1029/2018JC014661.

Kourafalou, V.H., L.-Y. Oey, J.D. Wang and T.N. Lee, 1996.  The fate of river discharge on the continental shelf. Part I: modeling the river plume and the inner-shelf coastal current. J. Geophys. Res, 101(C2), 3415-3434, doi: 10.1029/95JC03024. 

line 147 and 154. Why didn’t you use analysis or re-analsysis data (rather than forecast)?

We used forecast data because they have higher spatial and temporal (3 hours) resolution. The are updated twice daily and time 00 will be equal to the analysis.

line 162 This part could be made clearer. I assume you take ECMWF wave data to compute drift ; is the computation carried out by OpenOil? and what does OpenOil take as input? Not the full spectrum, I believe, so probably just the total SWH as I gather from the reference Breivik et al that you quote in the following. So, what is the use of the separation between Wind and Swell you mentioned earlier? CMWF WAM data do supply that information, but is it of any use to you?

The ECMWF wave model provides the surface Stokes drift, significant wave height, and mean wave period. Using the approximated Stokes drift profile of Breivik et al. 2016, we calculate vertical profiles of the Stokes drift. The directional difference between wind sea and swell is thereby not taken into account and we expect that the contribution from swell at depth is negligible as the oil is dominantly at or near the surface. What is taken into account is that older seas provide a deeper Stokes drift profile than young wind sea. A separation of surface oil and submerged oil in the presence of wind and older seas (or swell) can however take place because the surface slick is also subject to direct wind drag, which is not necessarily in the same direction as the dominant wave direction, particularly when the wind direction is changing quickly.

line 221 to 231

How is the computation of the wave drift actually carried out? See also remark above, at line 162 You mention a 2% wind speed velocity additional factor, beside the computed Stokes drift, to reproduce the observations. Is it your case too? Did you use this additional factor to fit your data? This is an important aspect, because, as your reference (Jones et al) quite rightly states, the mechanism behind this additional direct is not clear. Any information to this regard would be very useful to the readers.

We apply a 2% wind drag  factor to oil that is part of the surface slick. In addition, the Stokes drift is applied for oil both at the surface and submerged oil (however using Stokes drift for the respective depth, see previous point.) This calculation is not included in the revised manuscript, but thoroughly explained in Røhrs et al, 2018. The combined drift due to Stokes drift and waves can thereby be up to 3-4% of the wind speed, dependent on wave conditions. While the mechanisms behind the additional wind drag is not entirely clear, Laxague et al 2018 provide some observational evidence and explanation for it, i.e. lack of wind-induced shear in the upper centimeters for the ocean model data.

And have you considered also the possibility that dispersion (as opposed to net transport ) due to wave drift might play a role? Indeed the effects of wave drift not only determine the position of a spill but also influence its spreading due to the non to the angular spreading of the wave spectrum.

We thank the reviewer for bringing up this interesting issue. We do not have the means to evaluate this spreading from the given output of the wave model, as wave spectra are not stored with enough spatial and temporal resolution. This would also make OpenOil more computationally demanding and slower. However, this feature can be considered in future versions of the model system. This is now mentioned in the discussion section.

The following two papers might be useful:
Carratelli, E.P., Dentale, F., Reale, F. On the effects of wave-induced drift and dispersion in the deepwater horizon oil spill(2011) Geophysical Monograph Series, 195, pp. 197-204.Giarrusso, C.C., Pugliese Carratelli, E., Spulsi, G. On the effects of wave drift on the dispersion of floating pollutants (2001) Ocean Engineering, 28 (10), pp. 1339-1348.

These papers are very relevant to our work and are now properly cited in the discussion part of the manuscript.

line 232 what do you mean by “three components of the horizontal drift “? Can you explain?

We mean currents, Stokes drift and wind drift. This is now spelled out.

Round 2

Reviewer 1 Report

The quality of the manuscript has been significantly improved through revision. There are still some minor issues that should be clarified before it is accepted for publication:

(1) It is mentioned in the rebuttal letter that the abstract has been shortened considerably. However, the abstract is not seen in the revised manuscript.

(2) Line 88, this sentence is still not accurate -- Reference [25] is not about operational oil spill modeling at all.

(3) Reference [54] is wrong in publication year (should be 2011, not 2013) and the original publisher name (AGU/geopress, not Weily). The full citation is:

Liu, Y., A. MacFadyen, Z.-G. Ji, and R.H. Weisberg (Editors) (2011), Monitoring and Modeling theDeepwater Horizon Oil Spill: A Record-Breaking Enterprise, Geophys. Monogr. Ser., 195, 271 PP., AGU/geopress, Washington, D.C., USA

Author Response

The quality of the manuscript has been significantly improved through revision. There are still some minor issues that should be clarified before it is accepted for publication:

We thank the reviewer for these encouraging remarks. Below are our point to point responses.

(1) It is mentioned in the rebuttal letter that the abstract has been shortened considerably. However, the abstract is not seen in the revised manuscript.

We apologize for the missing abstract. Unfortunately we had some problems with formatting the latex/pdf of the revised manus. The formatting has been corrected now.

(2) Line 88, this sentence is still not accurate -- Reference [25] is not about operational oil spill modeling at all.

Dagestad 2018 should have been cited here, and this is now corrected. Li (2019) is cited elsewhere.

(3) Reference [54] is wrong in publication year (should be 2011, not 2013) and the original publisher name (AGU/geopress, not Weily). The full citation is:

Liu, Y., A. MacFadyen, Z.-G. Ji, and R.H. Weisberg (Editors) (2011), Monitoring and Modeling theDeepwater Horizon Oil Spill: A Record-Breaking Enterprise, Geophys. Monogr. Ser., 195, 271 PP., AGU/geopress, Washington, D.C., USA

We used the reference found by google scholar. We have now corrected this reference as suggested by the reviewer.